# Enhancing Immunotherapy in Ovarian Cancer: The Emerging Role of Metformin and Statins

**DOI:** 10.3390/ijms25010323

**Published:** 2023-12-25

**Authors:** Diana Luísa Almeida-Nunes, Ricardo Silvestre, Ricardo Jorge Dinis-Oliveira, Sara Ricardo

**Affiliations:** 1Differentiation and Cancer Group, Institute for Research and Innovation in Health (i3S) of the University of Porto, 4200-135 Porto, Portugal; dnunes@i3s.up.pt; 21H-TOXRUN—One Health Toxicology Research Unit, University Institute of Health Sciences, CESPU, CRL, 4585-116 Gandra, Portugal; ricardo.dinis@iucs.cespu.pt; 3Life and Health Sciences Research Institute (ICVS), School of Medicine, University of Minho, 4710-057 Braga, Portugal; ricardosilvestre@med.uminho.pt; 4ICVS/3B’s—PT Government Associate Laboratory, 4710-057 Braga, Portugal; 5UCIBIO-REQUIMTE, Laboratory of Toxicology, Department of Biological Sciences, Faculty of Pharmacy, University of Porto, 4169-007 Porto, Portugal; 6Department of Public Health and Forensic Sciences, and Medical Education, Faculty of Medicine, University of Porto, 4169-007 Porto, Portugal; 7FOREN—Forensic Science Experts, 1400-136 Lisboa, Portugal; 8Faculty of Medicine, University of Porto, 4169-007 Porto, Portugal

**Keywords:** ovarian cancer metabolomics, T-cell exhaustion, drug repurposing, clinical trials

## Abstract

Ovarian cancer metastization is accompanied by the development of malignant ascites, which are associated with poor prognosis. The acellular fraction of this ascitic fluid contains tumor-promoting soluble factors, bioactive lipids, cytokines, and extracellular vesicles, all of which communicate with the tumor cells within this peritoneal fluid. Metabolomic profiling of ovarian cancer ascites has revealed significant differences in the pathways of fatty acids, cholesterol, glucose, and insulin. The proteins involved in these pathways promote tumor growth, resistance to chemotherapy, and immune evasion. Unveiling the key role of this liquid tumor microenvironment is crucial for discovering more efficient treatment options. This review focuses on the cholesterol and insulin pathways in ovarian cancer, identifying statins and metformin as viable treatment options when combined with standard chemotherapy. These findings are supported by clinical trials showing improved overall survival with these combinations. Additionally, statins and metformin are associated with the reversal of T-cell exhaustion, positioning these drugs as potential combinatory strategies to improve immunotherapy outcomes in ovarian cancer patients.

## 1. Introduction

Ovarian cancer (OC), one of the most challenging gynecological malignancies, is often associated with the development of malignant ascites (MA)—an accumulation of fluid in the peritoneal cavity. This condition, a hallmark of peritoneal carcinomatosis [1], is present in over a third of OC patients at the time of initial diagnosis and is almost ubiquitous in cases of relapse [2,3,4]. The presence of MA is not only an indicator of advanced disease but also correlates with a worse prognosis [2]. Globally, OC stands as the deadliest gynecological cancer, with more than 125,000 women dying from this disease every year. Alarmingly, these numbers are predicted to rise by 67% by 2035, potentially leading to over 250,000 deaths [5]. The high mortality of OC is largely attributable to late diagnosis and the limited availability of targeted therapies [6]. OC displays multiple histological types and molecular subtypes that involve different origin cells and different patterns of progression and response to therapy [2,7,8]. The most common and aggressive subtype is high-grade serous carcinoma (HGSC), in which the presence of MA and advanced-stage disease are independent predictors of poor prognosis, often contributing to chemoresistance, metastasis, and decreased tumor resectability [3,4,9,10]. Patients with advanced OC typically undergo debulking surgery to remove the primary tumor and all metastatic implants. In cases where numerous tumor masses are present in the abdominal cavity at diagnosis, neoadjuvant chemotherapy is typically employed to reduce MA levels and minimize postoperative complications [11]. However, cytoreductive surgery, a complex procedure that involves multiple tissue extractions, may compromise the feasibility of future surgeries, especially in cases of relapse [11]. These relapsed tumors are frequently resistant to chemotherapy, necessitating repeated paracenteses to manage MA [12]. Since sampling of tumor cells from the primary site during OC progression is challenging, analysis of tumor cells is primarily conducted by centrifuging the ascitic fluid, which is drained from patients at different times during its clinical path, providing a unique window into the disease’s evolution [12].

The ascitic fluid contains a mixture of tumor and non-tumor cells, including fibroblasts, adipocytes, mesothelial, endothelial, and immunologic cells [13], as well as cell-free DNA and several signaling molecules that mediate cell behavior [1]. The pathogenesis of MA in OC is complex and multifactorial. It is generally accepted that MA occurs as a disruption in the balance between fluid production and reabsorption, facilitated by increased capillary permeability. This increase is primarily driven by the upregulation of vascular endothelial growth factor (VEGF) [14] or, in case of impaired lymphatic drainage of the abdomen, is due to the obstruction of lymphatic stomata by tumor cells [15,16].

The acellular fraction of this ascitic fluid contains tumor-promoting soluble factors, bioactive lipids, cytokines, and extracellular vesicles, which interact with the cells present in MA [17]. Numerous studies have demonstrated that the peritoneal environment in OC is advantageous to tumor cell proliferation and invasion, in contrast to its quiescent state in normal conditions [18]. The pro-inflammatory signature associated with OC promotes angiogenesis and exerts chemotactic and protective effects on cancer cells [13]. While tumor cells contribute to the secretion of angiogenesis-modulating factors, non-transformed tumor-infiltrating cells such as fibroblasts, myeloid cells, immune cells, and endothelial precursors also play a crucial role in modulating neo-vascularization [13]. The metabolome profiling of MA has revealed significant differences in pathways involving fatty acids, cholesterol, glucose, and insulin [1]. All of these factors in the MA microenvironment promote tumor growth, resistance to chemotherapy, and immune evasion [19], unveiling a crucial role of this serous fluid in OC progression [20].

In the following sections, we scrutinize the role of cholesterol and insulin pathways in OC, by revising established information as well as new insights obtained by recent work in this field. We systematically review the components of these metabolic pathways to provide insights into the process of tumor cell dissemination, and we conclude by suggesting that studying these pathways allows us to identify potential targets for intervention that may lead to improved patient outcomes.

## 2. The Impact of Cholesterol on Ovarian Cancer Metabolism

The tumor microenvironment (TME) of MA from OC differs from other cancer types, characterized by an adipocyte- and lipid-rich milieu, which has been shown to contribute to tumorigenesis, tumor immune evasion, chemoresistance, and cancer recurrence [21,22,23]. In this lipid-rich TME, ovarian tumor cells predominantly use lipid-dominant pathways and other alternative metabolic trails [24]. Co-culture studies of adipocytes and OC cells showed that adipocytes promote homing, migration, and invasion of tumors, mediated by adipokines (such as IL-8) [22]. A direct transfer of lipids from adipocytes to OC cells and induced lipolysis in adipocytes and β-oxidation in cancer cells has also been reported, suggesting that adipocytes act as an energy source for the cancer cells [22]. Nieman and colleagues proved that adipocytes offer a proliferative advantage by transferring fatty acids (FAs) to OC cells [22]. These interactions between adipocytes and OC cells result in metabolic alterations in both cell types, similar to what happens in interactions between adipose tissue and contracting muscle. The energy for contracting muscle is provided by FAs mobilized from adipocytes, and the transport of free FAs depends on the lipolysis of stored triglycerides to free FAs and glycerol. These studies proved that cancer cells induce adipocyte lipolysis [22,25]. Lipolysis and lipogenesis are important for maintaining high ATP production in OC cells, which is essential for proliferation [26]. Lipolysis also contributes to the high free FA content in ascitic fluid that may contribute to the metabolic reprogramming of OC cells from aerobic glycolysis to fatty acid β-oxidation (FAO), because of phosphorylation of AMP-activated protein kinase (AMPK) that favors energy production by β-oxidation. This metabolic switch occurs by phosphorylation of acetyl-CoA carboxylase (ACC), by AMPK or protein kinase A, resulting in its inactivation and incapacity to inhibit carnitine palmitoyltransferase 1 (CPT1). CPT1 is an important regulating enzyme of FAO in the form of acyl-CoA. The OC cells in the presence of adipocytes increased the phosphorylation of AMPK, the activity of protein kinase A, and the rate of β-oxidation, by increasing the levels of CPT1 and acyl-CoA oxidase 1, the first enzyme in the β-oxidation pathway, allowing OC cells to prosper on lipids acquired from adipocytes [22]. It has also been shown that OC cells use free FAs from MA or omentum-conditioned media and, in response, alter the expression of genes involved in FAO and lipogenesis [26]. Anchorage-independent cancer cells, a feature of cancer stem cells, can also switch metabolism and use FAs for β-oxidation as an alternative energy source, increasing the production of ATP and NADPH [27,28]. Multicellular aggregates found in MA are also frequently hypoxic, and hypoxic cells have elevated FA uptake, created by degradation of triglycerides present in lipid droplets [29]. Cholesterol and FAs are two main types of lipids. Cholesterol is a crucial metabolite for mammalian cells that helps to maintain the structural integrity and fluidity of the plasma membrane. It regulates cell-to-cell interactions by mediating cellular proliferation, immunity, and inflammation signaling pathways [30]. Several routes of cholesterol metabolism within cells have been described (Figure 1), including (a) de novo cholesterol synthesis, (b) exogenous cholesterol uptake, (c) cholesterol storage, (d) cholesterol conversion, and (e) cholesterol trafficking [31].

De novo cholesterol synthesis starts at acetyl-CoA via a complex enzymatic process [Figure 1a]. In these reactions, 3-hydroxy-3-methylglutaryl-CoA (HMG-CoA) reductase (HMGCR), farnesyl-diphosphate farnesyltransferase 1 (FDFT1), and squalene epoxidase (SQLE) are key rate-limiting enzymes that convert HMG-CoA to mevalonate and squalene to 2,3-epoxysqualene [31]. Circulating cholesterol enters human cells through the interaction between low-density lipoprotein (LDL) and LDL receptor (LDLR), which transports the cholesterol into cells by endocytosis [Figure 1b] [26]. However, free intracellular cholesterol levels require severe control within the cytoplasm because high levels lead to lipo-toxicity [30]. An increased free cholesterol concentration above 5% activates the binding of sterol regulatory element-binding protein (SREBP) cleavage-activating protein (SCAP) and Insig-1 on the endoplasmic reticulum (ER) membrane. This activation results in the retention of the SCAP-SREBP complex within the ER, thereby inhibiting cholesterol/FA synthesis and transportation and preventing lipid toxicity [32]. Sterol O-acyltransferase (SOAT) is allosterically activated by elevated intracellular free cholesterol levels, promoting the conversion of cholesterol to cholesterol-ester, which is stored in lipid droplets [Figure 1c] [33]. Oxysterol, a product of excess cholesterol, acts as a ligand that directly activates the liver X receptor (LXR) transcription factor. This activation regulates the cholesterol efflux pathway [Figure 1d], mediating the gene expression of the ATP-binding cassette (ABC) transporters, such as *ABCA1* and *ABCG1* [Figure 1e] [34,35]. The cholesterol exported by ABCA1 is transported by lipid-free apolipoprotein A-I, producing immature high-density lipoprotein (HDL) that is converted into mature HDL by lecithin-cholesterol acyltransferase (LCAT) in the plasma [36]. However, cholesterol exported by ABCG1 can directly become mature HDL [36], which can be consumed by OC cells by binding to HDL receptor-scavenger receptor type B1 (SR-B1), resulting in selective cholesterol-ester uptake for subsequent activation of downstream pathways involved in cancer cell proliferation, growth, and migration [37].

Previous studies have shown high cholesterol levels in MA of OC [27]. One such study by Helzlsouer and colleagues initially reported that the cholesterol concentration in the blood was proportionally correlated with the risk of OC [38]. Furthermore, high levels of LDL and cholesterol are associated with aggressive tumor biology (increased cellular proliferation and chemoresistance) and worse survival outcomes in OC patients [39]. In the murine ID8 OC model, mice subjected to a high-cholesterol diet displayed increased tumor growth compared to control groups [40]. Dysregulated cholesterol homeostasis has also been reported to increase platinum resistance in OC [41]. Additionally, high cholesterol levels in MA have been shown to contribute to cisplatin resistance in ovarian tumor cells by activating an LXR α/β nuclear receptor, with sequential upregulation of multidrug resistance protein 1 (MDR1), also known as P-glycoprotein (P-gp) [42]. However, the effects of dysregulated cholesterol homeostasis on the mitochondria of OC cells warrant further investigation. Elevated mitochondrial cholesterol levels can disrupt its function, inhibiting mitochondrial membrane permeabilization and releasing cytochrome c (pro-apoptotic signal), contributing to chemotherapy resistance [37].

Beyond its effects on tumor cells, cholesterol may contribute to immunosuppressive TME. For example, it has been shown that cholesterol affects tumor-associated macrophages (TAMs) present in MA, as tumor cells secreted high molecular weight hyaluronic acid that can alter macrophage membrane composition by significantly decreasing membrane cholesterol content. This alteration in membrane composition can change macrophage activation, promoting IL-4/PI3K/Akt/STAT6-mediated pro-tumor reprogramming [43]. Peritoneal resident macrophages also exhibit tumor cell-induced increase in FAO and production of itaconic acid, which increases oxidative phosphorylation-mediated ROS generation in macrophages and tumor cells [44]. A possible therapy could involve inhibiting cholesterol efflux in TAM cells to reverse the pro-tumorigenic effect. This could be achieved by deactivating the *ABCA1* or *ABCG1* genes, which are responsible for cholesterol transport, leading to lipotoxicity and, ultimately, tumor cell death [37]. Polyunsaturated FAs, such as prostaglandin E2, can also act as immune suppressors by inhibiting the expression of Th1 cytokines, such as tumor necrosis factor α (TNFα), interferon-gamma (IFNγ), and interleukin (IL)-2, while increasing the expression of the cytokines IL-4, IL-6, and IL-10 [45]. Lipoproteins (LPA), abundant in MA, can increase de novo lipid synthesis in OC and promote proliferation [27,46]. Both LPA and sphingosine-1-phosphate stimulate IL-8 expression by OC cells, contributing to the high cytokine levels in MA [47]. It is commonly accepted that obesity and adipocytes foster persistent inflammation characterized by increased cytokines, including IL-6, IL-8, and vascular endothelial growth factor (VEGF) [48], all of which are elevated in MA and contribute to reciprocal crosstalk. However, the use of anti-VEGF therapies as a frontline treatment for OC is limited, as OC cells can alter their lipid metabolism in response to these therapies, leading to alterations in the lipidomic profile associated with antiangiogenic drug resistance [48].

### The Therapeutic Potential of Statins in Ovarian Cancer Management

Statins are specific inhibitors of HMGCR that block the mevalonate pathway [49]. They were initially used to lower the cholesterol level in the blood and were found to be well-tolerated [49]. Jiangnan He and colleagues showed that the upregulation of mevalonate pathway proteins is primarily mediated by TP53 mutations, a common dominant genetic mutation in this cancer type [37]. Numerous studies confirmed that lipophilic statins, such as simvastatin and lovastatin, significantly reduced cell viability and proliferation, stemness, invasion, migration, and increased mitochondrial apoptosis and chemotherapeutic sensitivity of OC cell lines and primary OC samples derived from patients or mouse models without causing injury to normal cells [50,51,52,53,54]. Statins block drug efflux pumps by a mevalonate-independent mechanism [55]. Göbel and co-workers showed that lipophilic statins attenuated the expression of IL-6, IL-8, VEGF, and transforming growth factor beta (TGFβ), which contributed to ovarian tumor progression [56]. Treatment of OC cell lines with statins-activated c-Jun N-terminal kinase (JNK) signaling, induced the pro-apoptotic protein Bim, reduced c-Myc phosphorylation, and blocked Ras/Rho signaling [51,54,57]. Also, statins improve antigen presentation in dendritic cells (DC) and T cell cytotoxic functions in a B16 melanoma mouse model by decreasing Rab5 protein prenylation, which is involved in the endosomal trafficking process. This leads to reduced antigen internalization and degradation at the cell surface [58]. These efficacy results and a favorable safety profile demonstrate that statins are a valuable therapeutic option in OC management.

## 3. Insulin-like Growth Factor System in Ovarian Cancer

Insulin-like growth factors (IGFs) play a significant role in normal and tumor cells [59]. The IGF signaling pathway is made up of several components, including transmembrane insulin-like growth factor receptors type 1 (IGF-1R) and type 2 (IGF-2R), and insulin receptor (IR) -A and -B; growth factor ligands include IGF-1, IGF-2, and insulin; and IGF binding proteins (IGFBPs) that regulate the availability of cellular IGF-1 and IGF-2 [60]. This pathway regulates physiological and pathophysiological processes involved in glucose metabolism and cell proliferation [61,62,63]. IGF-1, IGF-2, and insulin bind to their respective receptors, which are part of the tyrosine kinase receptor family. When co-expressed in cells, IGF-1R forms hybrid receptors with a high affinity for IGF-1 whereas a lower affinity for insulin. The IGF system is vital for cell growth and survival, whereas insulin predominantly regulates cell metabolism [64]. IGF-2R requires an intracellular tyrosine kinase domain, so it binds exclusively to IGF-2 without affecting cell proliferation [65]. Activation of the IGF-signaling cascade occurs when the IGF-1 binds to IGF-1R (Figure 2), activating intracellular tyrosine kinase domains by autophosphorylation. This activation phosphorylates the insulin response substrate (IRS) 1 and IRS2, initiating a cascade involving adaptor proteins as SHC [66]. Tyrosine phosphorylation of the IRSs activates the phosphatidylinositol 3-kinase (PI3K) pathway and triggers several biological responses, while tyrosine phosphorylation of SHC induced downstream signaling activation through the Ras/Raf/MEK/Erk pathway [66,67]. PI3K converts phosphatidylinositol 3, 4 phosphate (PIP2) into phosphatidylinositol 3, 4, 5 phosphate (PIP3), which in turn activates Akt phosphorylation [66]. Tuberous sclerosis protein 1/2 is downstream of Akt and inhibits the mammalian target of rapamycin (mTOR) that regulates cell proliferation [66]. This pathway inhibits apoptosis through phosphorylation of BCL2-associated agonists of cell death (BAD) and FKHR. It can also activate SHC and GRB2 adaptor proteins, which activate the mitogen-activated protein kinase (MAPK) pathway, resulting in cell proliferation [68]. IGFBPs can inhibit nuclear factor-κB (NFκB) and promote cell death through caspase-8 activation [69]. In summary, the IGF-1R-signaling pathway promotes cell proliferation through the MAPK pathway and inhibits the apoptosis pathway by blocking proapoptotic proteins like BAD [70]. It also prevents apoptosis by phosphorylating apoptosis signal-regulating kinase 1 (ASK1), a key player in the TNF-α-induced apoptosis, thus ensuring cell survival under oxidative stress [71].

IGF-1R, IR-A, IR-B, IGF-1, and IGF-2, as well as their regulating IGFBPs, are expressed in OC cell lines and tissues [72,73,74,75,76]. IGF-1R is the main receptor expressed in OC cells, which favors IGF-mediated signaling over insulin-mediated signaling in these cells. In addition, *IGF-1R* gene expression correlates with cisplatin resistance [77,78]. Consequently, numerous therapeutic strategies have been developed to inhibit or prevent the activation of the IGF signaling pathway in cancer cells, primarily through IGF-1R blocking antibodies and tyrosine kinase inhibitors that impede the tyrosine kinase domains of IGF-1R and IR [79,80]. In preclinical studies, targeted strategies of anti-IGF-1R/IR effectively reduced OC cell growth and enhanced the efficacy of platinum-based chemotherapy [59]. However, promising preclinical data, these targeted strategies have shown limited efficacy in clinical applications, raising questions about the therapeutical potential of targeting the IGF-1R/IR signaling pathway.

On the other hand, *IGF-2R* gene expression is often reduced in OC, aligning with its potential role as a tumor suppressor. This reduction is associated with decreased cell proliferation, local invasion and metastasis, and increased apoptosis [75]. Both IGF-1 and IGF-2 have been implicated in OC progression, metastization, and chemotherapeutic response in patients [81,82,83,84]. The expression levels of IGF-1 and IGF-2 are significantly increased in OC tissues compared to their benign counterparts [82,85]. In addition, OC cells secrete IGF-1 and IGF-2, indicating autocrine and/or paracrine signaling [86,87]. High *IGF-1* gene expression in OC tumors has been associated with intrinsic resistance to platinum-based chemotherapy [81,88]. In OC cell lines, IGF-1 treatment induced cisplatin resistance via IGF-1R/PI3K pathway activation, whereas IGF-1R/PI3K inhibition re-sensitized these cells to cisplatin [77]. In addition, IGF-2 mRNA was upregulated in paclitaxel-resistant OC cell lines compared to sensitive cell lines. IGF-2 knockdown turns these cell lines sensitive to paclitaxel, indicating a role for IGF-2 in mediating paclitaxel resistance [84].

### Metformin as a Therapeutic Option to Blockade Insulin-like Growth Factor System

Metformin has demonstrated antitumor effects in several types of cancer in preclinical studies [89,90]. Although not consensual, some epidemiologic studies show that OC patients treated with metformin exhibit a significantly higher overall survival (OS) compared to those not taking metformin [91,92,93,94,95]. Several mechanisms are reported to support metformin’s anticancer activity, such as modulation of AMPK signaling, regulation of AKT activity, and activation of the apoptosis cascade [96,97]. Additional metabolic changes have been identified that are associated with processes like gluconeogenesis, mitochondrial activity, and cellular metabolism [98,99]. In addition, metformin has been reported to inhibit epithelial–mesenchymal transition (EMT), inhibit IGF signaling, and selectively suppress cancer stem-like cell (CSC) differentiation [100,101,102,103,104]. Particularly in OC, it was shown that metformin reverses chemotherapy resistance, reduces cancer cell migration and metastatic potential, and inhibits EMT [96,99,105,106,107]. In OC mice models, metformin treatment decreases IGF-1 levels [108] and the IGF-1 signaling pathway in uterine serous carcinoma [102]. The binding of IGF-1/IGF-1R activates AKT and ERK signaling in OC cells, which induces the activation of mTOR, which in turn controls protein translation and cell growth [109,110,111]. Some in vivo and in vitro studies also showed that metformin activates AMP-activated protein kinase (AMPK) [112,113], a heterotrimeric enzyme comprising one catalytic subunit (α) and two regulatory subunits (β and γ [114]). AMPK activation occurs in part by phosphorylation of the Thr172 residue of the α-subunit by two known upstream kinases: liver kinase B1 (LKB1) and calcium–calmodulin-dependent kinase kinase (CaMKK) [115,116]. The initial trigger of AMPK activation is a switch in the AMP/ATP ratio in response to several stimuli, including exercise [117], hypoxia [118], hormones [119,120] and drugs, including metformin [112]. Metformin-activated AMPK inhibits the mTOR signaling pathway in OC cells [106,121], reducing their proliferation and resistance to unfavorable conditions. This activation also decreases the signaling pathways mediated by AKT and ERK in OC cells [122,123,124]. These signaling pathways are associated with the increase of oncoproteins, such as the transcription factor c-MYC and the inhibitory apoptotic protein survivin (BIRC5) [124,125,126]. Metformin decreases c-MYC protein levels in OC cell lines [127,128] and plays a key role in regulating the metabolic plasticity of T cells, which is vital for T cell-mediated immunity. AMPK contributes to the development of memory T cells and promotes the CD8^+^ T cells to switch to fatty acid oxidation and/or mitochondrial oxidative phosphorylation, reducing their metabolic dependency on glycolysis and reactivating them for efficient antitumor immune responses [129]. The effects of metformin on cancer cells have sparked significant interest in the cancer research community to further explore its potential as a therapeutical option in cancer.

## 4. Clinical Trials Using Statins and Metformin in Ovarian Cancer Treatment

For the past 30 years, the standard treatment for OC has primarily been platinum-based therapies as first-line chemotherapy [130]. Although most OC patients initially respond to this upfront therapy, more than 75% of patients relapse due to the development of chemotherapy resistance [131]. This underscores the urgent need for novel therapeutic options to improve the OS of women with OC. Traditional drug discovery in oncology is costly and time-consuming, with a high failure rate in clinical trials due to issues such as high toxicity and/or lack of drug efficacy [132,133]. Drug repositioning has the potential to overcome these obstacles, offering a cost-effective and safe therapeutic approach leveraging the well-known toxicity profile of existing drugs [134]. To expedite drug discovery and identify potential new applications for existing non-cancer drugs [135,136], several epidemiological studies have explored new targets for non-cancer drugs, such as statins and metformin, for their potential as anticancer drugs in the OC context. Table 1 summarizes some of the clinical trials performed with these two repurposed drugs.

There are strong indications that patients who originally take statins have significantly higher OS rates than those who do not (hazard ratio, 0.71; 95% confidence interval: 0.63–0.80, *p* = 0.151) [137]. Also, even if statins are just taken after OC is diagnosed, the OS rate is significantly higher (hazard ratio, 0.87; 95% confidence interval: 0.80–0.95, *p* = 0.411) [138]. These results corroborate the two clinical trials, NCT04457089 with simvastatin and NCT00585052 with lovastatin, both associated with a lower risk of OC occurrence [139]. However, some prospective clinical trials of statins in cancer patients were unsuccessful in showing any survival benefit, partially attributed to the fact that the pharmacodynamic and pharmacokinetic factors were not considered [140]. Another aspect that could interfere with the efficacy is diet [52]. For example, pitavastatin can cause the regression of ovarian tumor xenografts in mice, but it is crucial to adapt the diet and eliminate geranylgeraniol supplement since this restores the growth of the xenografts in mice treated with pitavastatin [52]. Studies have shown that statins affect tumor growth in xenografts, but no tumor regression was observed nor dietary geranylgeraniol restriction was observed. Many human foods, especially oils [141], contain geranylgeraniol, and this can interfere with the anti-cancer activity of statins in patients, which could explain the failures of clinical trials with statins, as the diet is not controlled [140,142]. It is crucial to consider the statin-dosage in clinical trials is the one used for hypercholesterolemia treatment. High statin doses could potentially lead to myalgia and other unfavorable side effects [143], and these effects will have to be evaluated in future studies.

Similarly, statins and the use of metformin to treat OC have been supported by epidemiological data. In a study with type 2 diabetes patients treated with metformin when diagnosed with OC (n = 16 patients), compared with type 2 diabetes patients treated with another therapeutic agent (n = 28 patients), or OC patients without diabetes (not treated with metformin, n = 297 patients), it was found that the progression-free survival at five years was significantly better in OC patients taking metformin (51%, median 72 months; 95% confidence interval: 13.3–not estimable) than for those taking another diabetes agent (8%, median 10 months; 95% confidence interval: 13.3–37.2 months) or those without diabetes (23%, median 16 months; 95% confidence interval: 13.9–19.5 months, *p* = 0.03) [144]. Another study found that the 5-year survival rate (60 months) of patients treated with metformin (n = 72 patients, 67%) when diagnosed with OC was significantly higher compared with patients without metformin treatment (n = 143 patients, 47%), with hazard ratio 2.2; 95% confidence interval: 1.2–3.8, *p* = 0.007 [91]. Although these studies had only a few patients, this evidence supports the need to develop more clinical trials with more patients to clarify the effectiveness of metformin in OC treatment. So, at present, only four clinical trials are registered in the NIH ClinicalTrials.gov database to study metformin intake in association with carboplatin and paclitaxel in non-diabetic women with OC (NCT02437812, NCT02050009, NCT02122185, NCT01579812) from phase I to phase II of the studies. Clinical trial NCT01579812 by Brown and colleagues already has results demonstrating that tumors from the metformin-treated arm have a three-fold decrease in specific subpopulations of OC stem cells (ALDH^+^CD133^+^) with enhanced sensitivity to cisplatin ex vivo (this experiment was done with spheroids obtained from the cells of the patients) [145]. Metformin results were associated with better-than-expected OS, supporting its use in phase III of the study. Although metformin combined with carboplatin and paclitaxel is tolerable, the patients may present some side effects, such as diarrhea, hypomagnesemia, and myelosuppression [146]. However, it is important to conduct well-designed studies to evaluate the clinical benefits and adverse effects of this drug combination to accurately measure this association.

## 5. The Impact of Metformin and Statins in the Tumor Immune Microenvironment

It is well accepted that lipid metabolism regulates tumor cell growth and its metastatic capacity, but its crucial role in immune cells is starting to gain shape. In the MA microenvironment, polyunsaturated FAs mediate T-cell suppression [147], and high levels of cholesterol impair the cytotoxicity of CD8^+^ effector T cells and induce them in a state of exhaustion [148]. High cholesterol increases endoplasmic reticulum (ER) stress in CD8^+^ T cells and consequently results in XBP-1 activation (Figure 3), which promotes the expression of inhibitory checkpoint proteins, such as T cell immunoglobulin and mucin domain-containing protein 3 (TIM-3), programmed cell death protein 1 (PD-1), and lymphocyte activation gene 3 protein (LAG-3), and when they are ligated to its receptors, the CD8^+^ T cells enter in a state of exhaustion [148]. Curiously, Xia and colleagues demonstrated that the combination of statins with anti-PD1 antibodies generated an improved anti-tumor effect, unveiling a synergistic effect between statins and immunotherapy [58].

Regarding metformin, studies have shown both in vivo and in vitro that this antidiabetic drug activates AMPK in the cancer setting [112,113]. AMPK pathway is a metabolic key regulator for T cell-mediated immunity, so its activation induces PPAR-gamma coactivator 1α (PGC1α), which increases mitochondrial activity and synergistically suppresses tumor growth by phosphorylation of PD-L1 (Figure 3) [129]. The TME also presented a higher number of myeloid-derived suppressor cells (MDSCs), which inhibits immune reactivity while increasing the onset of MA formation and reducing patient survival [149,150]. The MDSCs represent a powerful mechanism of immunosuppression by the enzymatic activity of CD39 and CD73 [151]. Recent studies demonstrate that OC tumor cells with increased CD73 enzymatic activity can reduce T cell sensitivity [152] which means that MDSCs can directly suppress T cell function in the TME of OC. Li and colleagues proved that metformin treatment increased the production of granzyme B, perforin, and IFNγ by CD8^+^ T cells in vitro and in vivo studies, which was associated with decreased activity of immunosuppressive MDSCs resulting from CD39 and CD73 downregulation [153]. In addition, the level of expression of CD39 and CD73 in MDSCs, as well as CD8^+^ T cell function, were measured in patients with type 2 diabetes mellitus who take metformin [153]. In accordance with in vitro and in vivo results, metformin treatment significantly decreased CD39 and CD73 expression in MDSCs and increased CD8^+^ T cell function, measured by granzyme B production [153]. Intriguingly, these effects were observed in patients with diabetes before and after metformin treatment using a pairwise comparison (e.g., comparing the expression levels within patients following treatment) [153]. Presumably, these measurements were taken at the beginning and end of the 2-year prospective study; however, the exact duration and dosage of treatment are not specified [153]. Nonetheless, the results show that metformin treatment has a deep impact on MDSC behavior and indicate that this drug may be useful to revert TME immunosuppression [154].

## 6. Conclusions

When T cells are continuously exposed to an antigen, suppressive immune cells, or an inflammatory stimulus, they undergo repeated T cell receptor stimulation and progressively lose their ability to secrete IL-2, TNFα, and IFNγ. This process leads to a loss of their immune functions, a phenomenon known as immune exhaustion. It is widely accepted that an inadequate function of tumor-specific T cells is one of the mechanisms of resistance to Immune Checkpoint Inhibitors (ICI). This review compiles results from several studies indicating that both metformin and statins prevent T-cell exhaustion. To test the hypothesis that the use of pitavastatin or metformin could potentiate the effect of immunotherapy by reactivating the effector T-cell function, the development of functional assays to evaluate response and resistance to ICI therapy is crucial. Additionally, testing drug combinations capable of reversing this resistance is urgently needed to accurately model the native tumor immune microenvironment.

## Figures and Tables

**Figure 1 ijms-25-00323-f001:**
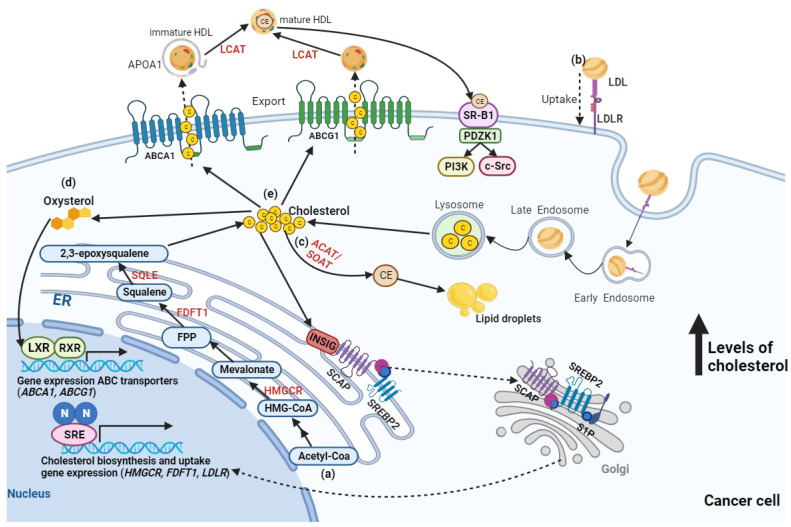
Metabolism of cholesterol within an ovarian cancer cell. (**a**) De novo cholesterol biosynthesis starts with acetyl-coenzyme A (ACoA) and is then synthesized in more than 20 enzymatic steps, while 3-hydroxy-3-methylglutaryl-CoA reductase (HMGCR), farnesyl-diphosphate farnesyltransferase 1 (FDFT1), and squalene epoxidase (SQLE) act as rate-limiting enzymes. (**b**) Cholesterol uptake is mediated by the ligation of low-density lipoprotein (LDL) to its receptor (LDLR), which is followed by endocytosis of LDL. This uptake creates high cholesterol accumulation, leading to intracellular lipo-toxicity and suppressing sterol-regulatory element binding protein 2 (SREBP2) transcription factor activity, thereby restricting the expression of enzymes involved in cholesterol synthesis and LDLR-mediated cholesterol uptake. (**c**) Excess cholesterol is converted into cholesterol esters (CE) by acyl-CoA: cholesterol acyltransferase 1 (ACAT1), also known as sterol-O-Acyl transferase 1 (SOAT1) enzyme, is then stored in lipid droplets. (**d**) Excess cholesterol is also converted to oxysterol through multiple enzymatic or non-enzymatic processes, which then activates liver X receptor (LXR)-retinoid X receptor (RXR) signaling and results in gene expression of ATP-binding cassette (ABC) subfamily A member 1 (*ABCA1*) and ABC subfamily G member 1 (*ABCG1*), which promote the (**e**) cholesterol efflux pathway. The cholesterol exported by ABCA1 is transported by lipid-free apolipoprotein A-1 (APOA1), producing immature high-density lipoprotein (HDL) that is converted into mature HDL by lecithin-cholesterol acyltransferase (LCAT) in the plasma. The cholesterol exported by ABCG1 can directly become mature HDL, which can be consumed by liver cells or steroidogenic cells (e.g., ovarian cells) by binding to HDL receptor-scavenger receptor type B1 (SR-B1), activating downstream pathways involved in cancer cell proliferation, growth, and migration. Figure created in BioRender.com (accessed on 24 November 2023). ER—Endoplasmic reticulum; FPP—Farnesyl pyrophosphate; INSIG—insulin-induced gene; PDZK1—PDZ Domain Containing 1; SCAP—SREBP-cleavage activating protein; S1P—site-1 protease; SRE—steroid response element.

**Figure 2 ijms-25-00323-f002:**
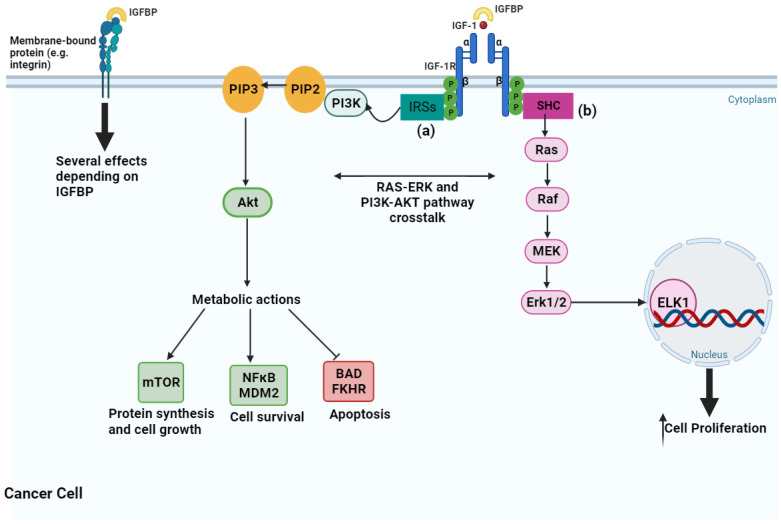
The insulin-like growth factor 1 (IGF1) signaling pathway and major downstream effects on cancer cells. The activation of IGF1 by the ligation to its receptor (IGF-1R) activates two pathways: phosphatidylinositol 3-kinase/Akt (PI3K/AKT) and Ras/mitogen-activated protein kinase (Ras/MAPK). (**a**) PI3K/AKT activates nuclear factor-κB (NFκB) and mouse double minute 2 (MDM2) for cell survival and inhibits apoptosis through inhibition of BCL2 associated agonist of cell death (BAD) and Forkhead transcription factor FOXO1 (FKHR), resulting in decreased apoptosis, increased protein synthesis, cell growth, and cell proliferation, among various other effects not represented here. (**b**) Ras/MAPK contains an elaborate kinase cascade that ultimately leads to increased cellular proliferation by promoting the activity of transcription factors, such as ELK1. The ligation of IGF-1 to IGF-1R is modulated by IGFBPs through direct binding in the extracellular space. IGFBPs also exert several IGF-independent effects via direct interaction with cell membrane-bound proteins, such as integrins. Image created in BioRender.com (accessed on 24 November 2023). Akt—Ak strain transforming; Erk—extracellular-signal-regulated kinase; ELK1—ETS Transcription Factor like-1; IGF-1—insulin-like growth factor 1; IGF-1R—insulin-like growth factor 1 receptor; IGFBP—insulin-like growth factor binding protein; IRSs—insulin receptor substrate proteins; MEK—mitogen-activated protein kinase; mTOR—mammalian target of rapamycin; P—phosphate; PI3K—phosphatidylinositol 3-kinase; PIP2—phosphatidylinositol 3, 4 phosphate; PIP3—phosphatidylinositol 3, 4, 5 phosphate; Raf—rapidly accelerated fibrosarcoma; Ras—rat sarcoma; SHC—Src homology/collagen.

**Figure 3 ijms-25-00323-f003:**
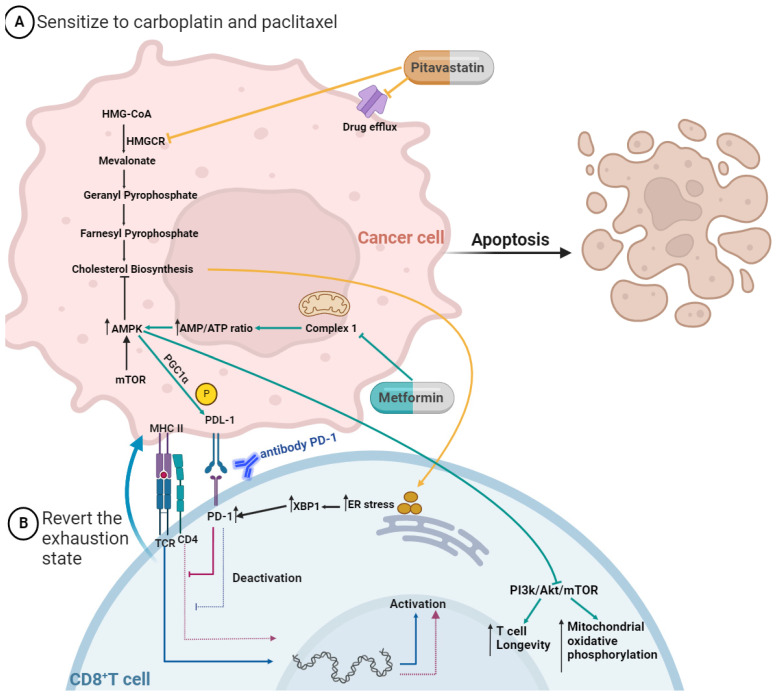
Mechanisms by which pitavastatin and metformin act to (**A**) turn cancer cells sensitive to chemotherapy and (**B**) help to revert the exhaustion state of CD8^+^ T cells. The pitavastatin blocks drug efflux pumps and inhibits HMGCR, leading to a cascade of inhibition, including cholesterol biosynthesis. The uptake of cholesterol by the exhausted CD8^+^ T cells increases the stress of the endoplasmic reticulum (ER), leading to the augmented expression of transcription factor X-box binding protein 1 (XBP1), which increases the expression of inhibitory receptors (PD-1). The metformin inhibits the respiratory-chain complex 1, which mediates the activation of adenosine monophosphate-activated protein kinase (AMPK), which inhibits the mammalian target of rapamycin (mTOR) and its downstream signaling pathways. This leads to increased T cell longevity and mitochondrial oxidative phosphorylation. AMPK activation induces PPAR-gamma coactivator 1α (PGC1α), which increases mitochondrial activity and synergistically suppresses tumor growth by phosphorylation of programmed cell death protein ligand-1 (PD-L1). CD8^+^ T cells are immune active when they have the ligation MHC II with TCR/CD4; on the contrary, when they express the receptor PD-1, they are immune inactive. So, for example, using an antibody to PD-1 allows the inhibition of ligation to PD-1 with PDL-1, and the CD8^+^ T cell stays active. Figure created in BioRender.com (accessed on 24th of November 2023).

**Table 1 ijms-25-00323-t001:** Clinical trials with statins and metformin with published results or ongoing for ovarian cancer treatment. Legend: I/II/III—clinical trial phase I/II/III; C—clinical trial completed; Re—a clinical trial in the phase of recruiting; S—clinical trial suspended to analyze the data.

				Clinical Trials					
Drug	Original Use	Mechanism of Action	Cancer Target	Title of Study and NCT Identifier	No. Patients	Objective	Therapeutic Scheme	Phase	Status/Year
Statins	To treat hyperlipidemia and prevent coronary artery disease, heart failure, and arrhythmia.	Lower blood cholesterol levels by blocking HMG-CoA reductase	HMG-CoA reductase and mevalonate pathway	A Phase II Study of the Synergistic Interaction of Lovastatin and Paclitaxel for Patients with Refractory or Relapsed Ovarian Cancer (NCT00585052)	11	Discover if the treatment combination of paclitaxel and lovastatin is more effective than the currently available chemotherapy for refractory or relapsed ovarian cancer.	Paclitaxel will be given at 80 mg/m^2^ IV over 1 h on day one and repeated weekly; lovastatin, 80 mg, orally, daily, will be self-administered by the subject.	II	C/2018
				Statin Therapy to Reduce Progression in Women with Platinum Sensitive Ovarian Cancer (NCT04457089)	20	Evaluate the possibility of using simvastatin intervention and estimate its effects on cancer progression.	Simvastatin 40 mg by mouth nightly for approximately six months during treatment with carboplatin and liposomal doxorubicin	Early Phase 1	Re/2023
Metformin	To treat type 2 diabetes	Promotes glucose transporter type 4 translocation to the plasma membrane that mediates the activation of liver kinase B1 and adenosine monophosphate-activated protein kinase (AMPK)	Inhibits the mitochondria respiratory chain, inhibiting mTOR	Phase Ib Study of Metformin in Combination with Carboplatin/Paclitaxel Chemotherapy in Patients with Advanced Ovarian Cancer (NCT02312661)	15	Single-center, dose-escalation trial with a traditional escalation rule with fixed dose levels (“3 + 3” rule). The recommended phase II dose will be defined as the maximum predefined dose level at which 0 of 3 or ≤1 of 6 subjects experience drug-related dose-limiting toxicity (DLT) during cycles 1 and 2 of treatment.	Metformin in increasing doses will be added to carboplatin/paclitaxel chemotherapy.	Ib	C/2018
				A Phase II Evaluation of Metformin, Targeting Cancer Stem Cells for the Prevention of Relapse in Patients with Stage IIC/III/IV Ovarian, Fallopian Tube, and Primary Peritoneal Cancer (NCT01579812)	90	The main goal is to determine if metformin improves the recurrence-free survival (RFS) of patients in relation to historical controls. Secondary objectives are: (a) compare the amount of cancer stem cells (CSC) in primary tumor specimens in metformin-treated patients versus matched controls, (b) determine if metformin improves overall survival relative to historical controls, (c) confirm the safety of metformin in non-diabetic ovarian cancer patients, and (d) correlate response rates with p53 mutations status, since metformin is mostly active in p53 mutant cells and p53 is mutated in ~50% of ovarian cancers.	Patients receiving primary surgical debulking followed by adjuvant chemotherapy will initiate metformin prior to primary surgery. Following surgery, patients will be initiated on metformin prior to the initiation of chemotherapy. Patients treated with neoadjuvant chemotherapy will initiate metformin treatment prior to the initiation of chemotherapy. Following surgery, patients will initiate metformin prior to the re-initiation of chemotherapy. The doses of metformin are 500 mg twice daily for seven days and then increased to 1000 mg twice daily. The chemotherapy administrated are carboplatin (AUC = 6) and paclitaxel (175 mg/m^2^) or carboplatin (AUC = 6) and taxol (80 mg/m^2^)	II	C/2018
				A Phase II, Open-Label, Non-Randomized, Pilot Study of Paclitaxel, Carboplatin, and Oral Metformin for Patients Newly Diagnosed with Stage II-IV Epithelial Ovarian, Fallopian Tube or Primary Peritoneal Carcinoma (NCT02437812)	30	A pilot study evaluating the safety, toxicity, and progression-free survival of advanced-stage ovarian carcinoma patients who underwent treatment with paclitaxel, carboplatin, and metformin.	Metformin (850 mg), Carboplatin (AUC 5 or 6), and Paclitaxel (80 mg/m^2^).	II	Re/2017
				A Randomized Placebo-Controlled Phase II Trial of Metformin in Conjunction with Chemotherapy Followed by Metformin Maintenance Therapy in Advanced Stage Ovarian, Fallopian Tube, and Primary Peritoneal Cancer (NCT02122185)	160	Determine if the addition of metformin to standard adjuvant or neoadjuvant chemotherapy plus extended metformin (metformin hydrochloride) beyond standard chemotherapy increases progression-free survival when compared to 6 cycles of standard chemotherapy alone in nondiabetic subjects with stage III or stage IV ovarian, primary peritoneal, or fallopian tube carcinoma. Evaluate metformin’s molecular mechanism of action in ovarian, fallopian tube, or primary peritoneal cancer by determining whether metformin’s anti-cancer effects are mediated by systemic metabolic changes and/or a direct effect on tumor cells, testing the metabolic and proteomic alterations induced in biospecimens from non-diabetic patients prospectively treated with standard chemotherapy in conjunction with metformin compared to placebo.	Patients receive a standard chemotherapy regimen which includes either paclitaxel intravenously (IV) over 2–3 h and carboplatin IV over 30–60 min on day 1; docetaxel IV over 1 h and carboplatin IV over 30–60 min on day 1; or paclitaxel IV over 1 h on days 1, 8, and 15, and carboplatin IV over 30–60 min on day 1. Treatment repeats every 21 days for up to 6 courses. Patients are randomized to condition 1 (metformin hydrochloride orally twice daily and standard chemotherapy regimen as above for 6 courses) or 2 (placebo orally twice daily and standard chemotherapy regimen as above for 6 courses). Treatment for metformin hydrochloride and placebo continues for up to 2 years in the absence of disease progression or unacceptable toxicity, and after completion of study treatment, patients are followed up for 2 years.	II	S/2023

## Data Availability

The authors confirm that the materials included in this chapter do not violate copyright laws. Where relevant, appropriate permissions have been obtained from the original copyright holder(s), and all original sources have been appropriately acknowledged or referenced.

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
