# Peer review of "Enhancing Immunotherapy in Ovarian Cancer: The Emerging Role of Metformin and Statins"

_ijms, 2023, doi:10.3390/ijms25010323_

Round 1

Reviewer 1 Report

Comments and Suggestions for Authors

The manuscript is a well written review on the effect of cholesterol and insulin pathways in ovarian cancer, highlighting the synergy of statins and metformin in antitumor therapy through the effect on T-cell exhaustion and on the immunosuppressive effect on myeloid-derived suppressor cells present in the tumor microenvironment.

They support their view with references to clinical trials and illustrate it all with nice figures.

Author Response

Dear Reviewer,

We appreciate your feedback and acknowledgement of the high quality of our review article on ovarian cancer treatment. The results of this bibliography search suggest that the administration of metformin and statins could significantly enhance the efficacy of immunotherapy by overcoming resistance to it. These findings have significant implications for the development of novel treatment strategies to treat ovarian cancer patients.

Sincerely yours,

Sara Ricardo

Reviewer 2 Report

Comments and Suggestions for Authors

This is a well performed and well written publication.

The manuscript focus  ovarian cancer metastization, which leads to malignant ascites, linked to poor prognosis. The acellular fraction of ascitic fluid contains tumor-promoting factors, bioactive lipids, cytokines, and extracellular vesicles, influencing peritoneal tumor cells. Understanding the liquid tumor microenvironment is crucial for developing efficient treatments.
This review focuses on ovarian cancer's cholesterol and insulin pathways, highlighting statins and metformin as effective treatments, supported by improved overall survival in clinical trials. These drugs also show promise in reversing T-cell exhaustion, suggesting potential combinatory strategies to enhance immunotherapy outcomes in ovarian cancer patients.

The Authors presented the clear introduction, well-planned publication.

The Discussion makes a reasonable effort to draw attention to the potential  clinical relevance of immune exhaustion. This phenomenon contributes to resistance to Immune Checkpoint Inhibitors (ICI) by impairing tumor-specific T cell function. Multiple studies suggest that both metformin and statins can prevent T-cell exhaustion.
The review emphasizes the importance of developing functional assays to assess response and resistance to ICI therapy, specifically investigating the potential of pitavastatin and metformin to reactivate effector T-cell function. Urgent exploration of drug combinations capable of reversing resistance is essential for accurately modeling the native tumor immune microenvironment. 

Author Response

Dear reviewer,

We appreciate your feedback and acknowledgement of the high quality of our review article on ovarian cancer treatment topic. The results of this bibliography search suggest that the administration of metformin and statins could significantly enhance the efficacy of immunotherapy by overcoming resistance to it. These findings have significant implications for the development of novel treatment strategies to treat ovarian cancer patients.

Sincerely yours,

Sara Ricardo

Reviewer 3 Report

Comments and Suggestions for Authors

1.   Is there any contra-indication or side effects of metformin and statins in  ovarian cancer combining treatment both in clinical or bench site? 

2.    In lines 373-376, and 394-399, it will be better to present the actual months of progression free survival and overall survival, hazard ration, 95% confidence interval and p value in details.  

Author Response

Dear Reviewer,

We appreciate your feedback and acknowledgement of the high quality of our review article on the topic of ovarian cancer treatment.

We also acknowledge the suggestions for improving its scientific soundness by adding the suggested information in the manuscript text:

  1. Contra-indications and side effects of the combination of metformin and statins with chemotherapy in ovarian cancer:

Page 13,  Lines 389-392: “It is crucial to consider the statin dosage in clinical trials is the one used for hypercholesterolemia treatment. High statin doses could potentially lead to myalgia and other unfavourable side effects[144], and these effects will have to be evaluated in future studies.”

Page 13, Lines 416-420: “Although metformin combined with carboplatin and paclitaxel is tolerable, the patients may present some side effects, such as diarrhoea, hypomagnesemia, and myelosuppression [148]. However, it is important to conduct well-designed studies to evaluate the clinical benefits and adverse effects of this drug combination to accurately measure this association.”

  1. We also included details about months of progression-free survival and overall survival, hazard ratio, 95% confidence interval and p-value.

Page 13, Lines 374-375 : “(hazard ratio, 0,71; 95% confidence interval: 0,63–0,80, P = 0.151)”

Page 13, Line 376 : “(hazard ratio, 0,87; 95% confidence interval: 0,80–0,95, P = 0.411)”

Page 13, Lines 398-399 : “(51%, median 72 months; 95% confidence interval: 13.3-not estimable)”

Page 13, Line 400 : “(8%, median 10 months; 95% confidence interval: 13.3–37.2 months)”

Page 13, Line 401 : “(23%, median 16 months; 95% confidence interval: 13.9–19.5 months, P = 0.03)”

Page 13 Lines 402-405 : “Another study found that the 5-year survival rate (60 months) of patients treated with metformin (n=72 patients, 67%) when diagnosed with OC was significantly higher compared with patients without metformin treatment (n=143 patients, 47%), with hazard ratio 2,2; 95% confidence interval: 1,2 – 3,8, p = 0.007”

We hope that we have addressed all of the issues that were suggested.

Sincerely yours,

Sara Ricardo